# In Situ Flow Cytometer Calibration and Single-Molecule Resolution via Quantum Measurement

**DOI:** 10.3390/s22031136

**Published:** 2022-02-02

**Authors:** Javier Sabines-Chesterking, Ivan A. Burenkov, Sergey V. Polyakov

**Affiliations:** 1Joint Quantum Institute, University of Maryland, College Park, MD 20742, USA; jsabines@umd.edu (J.S.-C.); ivan.burenkov@gmail.com (I.A.B.); 2National Institute of Standards and Technology, Gaithersburg, MD 20899, USA; 3Physics Department, University of Maryland, College Park, MD 20742, USA

**Keywords:** flow cytometry, single-molecule, single-photon sources

## Abstract

Fluorescent biomarkers are used to detect target molecules within inhomogeneous populations of cells. When these biomarkers are found in trace amounts it becomes extremely challenging to detect their presence in a flow cytometer. Here, we present a framework to draw a detection baseline for single emitters and enable absolute calibration of a flow cytometer based on quantum measurements. We used single-photon detection and found the second-order autocorrelation function of fluorescent light. We computed the success of rare-event detection for different signal-to-noise ratios (SNR). We showed high-accuracy identification of the events with occurrence rates below $10^{-5}$ even at modest SNR levels, enabling early disease diagnostics and post-disease monitoring.

## 1. Introduction

Flow cytometry is a commonly used optical technique to measure a wide range of cell properties in a high throughput manner by probing them with a laser as they flow through a flow chamber [1,2]. By introducing fluorescent markers to a sample, a flow cytometer can be used to detect the presence of target molecules and reveal the distribution of cells containing these molecules within a heterogeneous population [3,4]. There are certain situations where the target molecules are presented in trace amounts within the sample. In the case when only one or just a few fluorescent biomarkers are present in the probe volume of a flow cytometer, fluctuations in the laser power, optical loss, detector noise, and other factors make it hard to detect the fluorescent signal [5]. Although detection of single fluorescent molecules in flow cytometers has been previously claimed [6,7,8,9,10], these claims have not been rigorously proven using the quantum-mechanical formalism of the second-order coherence [11], a method that describes intensity correlations of light sources commonly measured using a Hanbury Brown and Twiss interferometer [12]. Previous claims rely on a priori knowledge of sample concentration and statistical analysis of photon-bursts, which do not necessarily arise from a single-emitter as large scatterers including bubbles within the flow cell could also generate these bursts. To this end, we present a theoretical model for sensing fluorescent biomarkers at the single-molecule level based on the measurement of the second-order autocorrelation function of their emitted light. This method provides in situ verification of single-molecule sensitivity from first principles. It can unambiguously resolve the number of emitters within the probe volume, even when background noise is present. The advantage of this method is that it creates an independent scale for resolving the number of particles based on laws of quantum mechanics [13,14,15]. This scale is unaffected by loss, and thus, this measurement enables an absolute calibration method for flow cytometers, which otherwise is a challenging task [16].

Using this scale we quantify the ability of our method to identify the presence or absence of a biomarker when its occurrence, *P*, in a sample population is extremely rare ($P<10^{-4}$). This level of sensitivity makes our scheme suitable for the diagnosis of early-stage diseases or the monitoring of disease recurrence. We show that identification of rarely occurring single biomarkers can be performed with a high success rate for achievable signal-to-noise ratios (SNR) values. We demonstrate that our method can be used to discriminate between different numbers of biomarkers, also with a high success rate.

## 2. Method

A source of light can be characterized using the second-order autocorrelation function. This function relates the photon number fluctuations at time *t* with those at a different time t+τ. In quantum mechanics, the second-order autocorrelation function is expressed in terms of creation ($a^{†}$) and annihilation (*a*) operators as:(1)$$g^{\left(2\right)}\left(\tau \right)=\frac{\langle a^{†}\left(t\right)a^{†}\left(t+\tau \right)a\left(t+\tau \right)a\left(t\right)\rangle }{{\langle a^{†}\left(t\right)a\left(t\right)\rangle }^{2}}.$$

When τ=0, Equation (Equation 1) gives
(2)$$g^{\left(2\right)}\left(0\right)=\frac{\langle \hat{n}\left(t\right)\left(\hat{n}\left(t\right)-1\right))\rangle }{{\langle \hat{n}\left(t\right)\rangle }^{2}},$$
where $\hat{n}$ is the photon number operator. It turns out that if light is generated by a single emitter, Equation (Equation 2) gives $g^{\left(2\right)}\left(0\right)=0$, because a single-photon emitter cannot emit more than one photon at a time. Note that for *N* single-photon emitters, the autocorrelation function at zero takes the following form:(3)$$g^{\left(2\right)}\left(0\right)=\frac{N-1}{N}.$$

The phenomenon that $g^{\left(2\right)}\left(0\right)<1$ for all N<∞ is known as antibunching. The dependence of the autocorrelation function $g^{\left(2\right)}\left(0\right)$ on the number of single-photon emitters is particularly strong for low *N*. Additionally, note from Equation (Equation 3) that $g^{\left(2\right)}\left(0\right)$ does not depend on optical loss. Therefore, the measurement of $g^{\left(2\right)}\left(0\right)$ allows quantification of the number of molecules present in the probe volume from first principles of optical quantum measurement.

It has been shown that we can infer the number of single-photon emitters within the collection volume based on the autocorrelation function and that a value of $g^{\left(2\right)}\left(0\right)<0.5$ implies that light comes predominantly from one single-photon emitter [11,13,17]. The relation between the autocorrelation function and the number of emitters has been studied widely and reported in numerous publications [18,19,20,21,22]. Here, we consider realistic experimental scenarios where the biomarker signal is polluted by noise. Typically, this noise is associated with the strong pump light used to excite fluorescence. Thus, it is common to encounter experiments plagued with optical background noise, such as thermal background, Raman scattering, autofluorescence, phosphorescence, scattered and unfiltered pump light, and ambient light leakage, among others. In addition, other effects, such as detector dark counts also contribute to background noise. In a faint coherent light field where $P_{1}\gg P_{2}\gg P_{3}\ldots$, the probability $P_{N_{\mathrm{ph}}}$ to observe $N_{\mathrm{ph}}$ photons quickly decreases with the number of detected photons. Using this assumption, we can use the approximate expression $g^{\left(2\right)}\left(0\right)\approx 2P_{2}/P_{1}^{2}$, where $P_{1}\approx Np_{b}+p_{n}$ and $P_{2}\approx p_{n}^{2}/2+Np_{n}p_{b}+N^{2}p_{b}^{2}/2\left(1-1/N\right)$ correspond to the probabilities of detecting one and two photons, respectively. Here, by introducing $p_{n}$ and $p_{b}$, the probabilities of detecting one photon from a noise source or from a single biomarker, respectively, we account for the presence of noise relative to the biomarker signal [23,24]. Substituting $P_{1}$, $P_{2}$ into the approximate expression for $g^{\left(2\right)}\left(0\right)$, we get:(4)$$g^{\left(2\right)}{\left(0\right)|}_{N}\approx \frac{N^{2}p_{b}^{2}\left(1-\frac{1}{N}\right)+2Np_{n}p_{b}+p_{n}^{2}}{{\left(Np_{b}+p_{n}\right)}^{2}}.$$

One can then harness this relation as a tool to connect the number of emitters to the observed autocorrelation values in a measurement device from first principles. By comparing the measured $g^{\left(2\right)}\left(0\right)$ with those for a known number of emitters we can find $p_{b}$ and $p_{n}$ specific to an experimental setup. Note that this approach requires a constant number of emitters within the collection volume throughout the calibration measurement. In the case of flow cytometry, the number of emitters is changing stochastically. We found an expression for an average number of biomarkers in the collection volume instead. For constant concentration, the number of biomarkers is governed by a Poisson distribution. Thus, for an average of 〈N〉 biomarkers in the interrogation volume we find:(5)$$g^{\left(2\right)}{\left(0\right)|}_{\langle N\rangle }\approx \frac{2\langle P_{2}\rangle }{\langle P_{1}^{2}\rangle }\approx \frac{\sum_{N=0}^{\infty }\left[\frac{e^{-\langle N\rangle }{\langle N\rangle }^{N}}{N!}\right]\left(N^{2}p_{b}^{2}\left(1-\frac{1}{N}\right)+2Np_{n}p_{b}+p_{n}^{2}\right)}{\sum_{N=0}^{\infty }\left[\frac{e^{-\langle N\rangle }{\langle N\rangle }^{N}}{N!}\right]{\left(Np_{b}+p_{n}\right)}^{2}},$$
where the summations can be done analytically:(6)$$g^{\left(2\right)}{\left(0\right)|}_{\langle N\rangle }\approx {\left[1+\frac{\langle N\rangle p_{b}^{2}}{{\left(\langle N\rangle p_{b}+p_{n}\right)}^{2}}\right]}^{-1}.$$

To understand this relation, consider the shape of the $g^{\left(2\right)}\left(\tau \right)$, Figure 1a. In general, any light field whose intensity fluctuates in time exhibits a shoulder of higher values for $0<\tau \ll t_{\mathrm{ToF}}$, where $t_{\mathrm{ToF}}$ is the characteristic time of intensity fluctuations; specifically, in flow cytometry, $t_{\mathrm{ToF}}$ is the time of flight of a particle through the interrogation volume. At $\tau \gg t_{\mathrm{ToF}}$
$g^{\left(2\right)}\left(\tau \right)$ reaches the uncorrelated background. If an assumption is made that light is only emitted by single-photon emitters, $g_{\mathrm{shoulder}}^{\left(2\right)}=1+1/\langle N\rangle$, (cf. [25]). We point out that the existence of the shoulder cannot be used for concentration estimation except when an additional assumption is made, because the shoulder can be due to any classical fluctuations. To prove the quantum character of the emitted light without extra assumptions, the difference between $g^{\left(2\right)}\left(0\right)$ and $g_{\mathrm{shoulder}}^{\left(2\right)}$ should be used. Therefore, for the dynamic measurements, we normalized the autocorrelation function $g^{\left(2\right)}\left(0\right)$ using the value $g_{\mathrm{shoulder}}^{\left(2\right)}$ [14], while in the static model, normalization at τ→∞ is ordinarily used.

In Figure 1b we show the behavior of $g^{\left(2\right)}\left(0\right)$ as a function of the number of emitters for both static (fixed number of emitters) and dynamic (varying but averaged number of emitters) cases for different values of the signal-to-noise ratio (SNR), defined here as $p_{b}/p_{n}$, i.e., SNR for a single emitter. As expected, the Poisson distribution correction for the dynamic case is most evident at low concentrations, i.e., precisely where $g^{\left(2\right)}\left(0\right)$ exhibits the most antibunching behavior, and therefore the averaging cannot be ignored.

Figure 1b shows that it is still possible to observe the relation between $g^{\left(2\right)}\left(0\right)$ and the average number of emitters even when the number of emitters changes at random during the measurement. Therefore, correlation-based absolute calibration of flow cytometers is possible. To calibrate, one needs to prepare a sample with biomarkers that are sufficiently diluted but whose exact concentration could be unknown. Then, a series of known dilutions should be applied to change the average number of emitters in the collection volume, and a series of $g^{\left(2\right)}\left(0\right)$ measurements should be taken. The ratio $p_{b}/p_{n}$, defined as SNR above, then can be extracted from the fit of the experimental data with Equation (Equation 5). Further, the absolute concentration of the biomarkers (i.e., the accurate scaling for horizontal axis in Figure 1b follows from the same fit.

In addition, from the coincidence analysis above, the rate of single counts provides information about the relation between $p_{b}$, $p_{n}$: $\langle P_{1}\rangle =\langle N\rangle p_{b}+p_{n}$, where 〈N〉 is the average number of biomarkers per measurement volume. This allows extraction of both $p_{b}$ and $p_{n}$ from the measurement. Then, based on the photon rate of a single emitter and the values of $p_{b}$, $p_{n}$, it is possible to create the absolute photonic flux (intensity) scale that is proportional to the number of emitters (biomarkers) in a particular biological sample, i.e., to perform absolute calibration of a flow cytometer.

It is evident that the $g^{\left(2\right)}\left(0\right)$ values flatten and asymptote towards 1 for a low SNR as seen in Figure 1b. Hence, the accuracy with which we can measure the concentration is limited by the SNR of a system. We point out that the absolute metrological measurements are possible for SNR ≳0.5, i.e., in the presence of relatively strong noise.

## 3. Proposed Experimental Setup

The proposed experimental setup is shown in Figure 2. A dilute sample labeled with fluorescent biomarkers flows through a flow cell where it is interrogated with a laser that excites the emitters. Then, the fluorescent emission is collected and spectrally filtered to remove noise produced by the bright excitation beam. Spectral filtering can be achieved using dichroic mirrors and low-pass filters (LP filters). The fluorescent emission is then split into two spatial channels using a 50:50 beam splitter. In this way, a Hanbury Brown–Twiss measurement can be performed using a pair of single-photon detectors and a time tagger [12]. The second-order autocorrelation function can be obtained by analyzing the coincidence counts between the detectors. Typically, to define a coincidence, delays between the detections on both detectors are measured, and a coincidence occurs if a delay is within a pre-defined interval (time bin), which allows calculation of the autocorrelation function. If excitation with short pulses is used, the predefined interval is typically equal to the inverse of the pulse repetition rate.

We estimated the expected number of photons in one measurement trial as $\lambda \approx M\cdot (Np_{b}+p_{n})$ where *M* is the number of recorded time bins during the ToF of a biomarker. The expression for λ can be rewritten using the average number of photons expected in one such measurement: $\lambda =N\langle N_{b}\rangle +\langle N_{n}\rangle$. In flow cytometry, the measurement trial duration is related to the velocity of the flow. To find typical parameters for this quantum measurement-limited flow cytometer, we considered standard operational settings. To estimate $\langle N_{n,b}\rangle$, we need to find the ToF of the biological entity with the biomarker through the collection volume. We considered a sample flow rate of $v_{s}=$ 10 μL/min and a sheath flow of $v_{sh}=$ 100 μL/min, a flow cell with cross-sectional area *A* = 250 μm × 250 μm, with an excitation beam waist σ≈ 1 μm. With these parameters the expected time of flight would be $t_{\mathrm{ToF}}=\sigma A/\left(v_{sh}+v_{s}\right)\approx$ 30 μs. As an excitation pump, we assumed a pulsed laser with a repetition rate of R= 80 MHz. We expected a biomarker with a quantum yield $Q_{y}\approx 10$% [26]. Given the ToF and the repetition rate, we can estimate the collection efficiency. For an objective with NA = 0.65, the one-half angular aperture is $\alpha =40^{\circ }$. The typical single-photon detector efficiency is η≈50%. Using these settings and estimated coupling loss $\eta_{c}=50$%, we can expect the average number of detected photons $\langle N_{b}\rangle \approx t_{\mathrm{ToF}}RQ_{y}\eta \eta_{c}\left(1-\cos\left(\alpha \right)\right)/2\approx 8$ per trial. On the other hand, the number of background photons is dependent on multiple factors unique to each setup, and it is harder to estimate. Raman scattering is dependent on the buffer solution, and our ability to filter it out depends on the absorption and emission spectra of the biomarker and the filters used. Using this estimation, here we took $\langle N_{b}\rangle$ = 8 and changed the value of $\langle N_{n}\rangle$ to obtain SNR = {0.5,1,2,4}.

## 4. Detection of Single Biomarkers

The capacity to detect a single emitter in flow cytometry can be an extremely advantageous tool for the diagnosis of early-stage diseases or the monitoring of a disease recurrence. Particularly important is detecting single rarely occurring biomarkers, for example, when samples have an occurrence rate (or the ratio of events with a biomarker present to the events without one) on the order of 1 in 1000 or less [27].

The identifying parameter is the number of photons detected during a single measurement trial, i.e., during the ToF of a biomarker through the collection volume. The likelihood that $N_{\mathrm{ph}}$ photons are detected during a trial given that the ensemble under measurement contains *N* emitters is governed by the Poissonian process: $L\left(N_{\mathrm{ph}}|N\right)=\lambda^{N_{\mathrm{ph}}}\exp\left(-\lambda \right)/N_{\mathrm{ph}}!$, where λ is the expected number of photons detected in one trial, and it depends on the flow cytometer setup—see the previous section.

In Figure 3 we present a panel of graphs that show the probabilities $L(N_{\mathrm{ph}}|N)$ to detect $N_{\mathrm{ph}}$ photons for ensembles of *N* emitters for different signal and noise levels. Here we were interested in a very low number of photons detected, as is usually the case when the number of biomarkers is low while the flow cytometer throughput is maximized.

From Figure 3 we see that those probability distributions become more distinct for higher SNR (lower noise levels) and can be used to distinguish between different numbers of emitters. It is important to distinguish between those distributions (Bayesian likelihoods) and probabilities of successful detection of a biomarker, particularly when biomarkers are rarely occurring in a sample.

The probability to identify a biomarker can be found using the likelihood and the expected occurrence rate using Bayes formula:(7)$$P\left(N|N_{\mathrm{ph}}\right)=\frac{L\left(N_{\mathrm{ph}}|N\right)P_{N}}{P_{N_{\mathrm{ph}}}},$$
where $P_{N}$ is the prior (previously known) probability to encounter an ensemble of *N* emitters in the sample, and $P_{N_{\mathrm{ph}}}$ is the probability to detect $N_{\mathrm{ph}}$ photons, which is given by the sum of likelihoods $L(N_{\mathrm{ph}}|i)$ for all possible number of emitters *i* multiplied by the prior probabilities $P_{i}$:(8)$$P\left(N_{\mathrm{ph}}\right)=\sum_{i=0}^{\infty }L\left(N_{\mathrm{ph}}|i\right)P_{i}.$$

Here we focus on the ultimate goal of quantum-enabled flow cytometry—the distinguishing of volumes with one emitter from those with no emitters assuming a low rate of occurrence. Under this condition, the probability of encountering a trial with more than one emitter can be neglected. We were interested in identifying the fraction of occasions when the positively identified volume indeed contains a single emitter:(9)$$S\left(N_{\mathrm{ph}}^{*}\right)=\frac{\sum_{N_{\mathrm{ph}}=N_{\mathrm{ph}}^{*}}^{\infty }L\left(N_{\mathrm{ph}}|1\right)P_{1}}{\sum_{N_{\mathrm{ph}}=N_{\mathrm{ph}}^{*}}^{\infty }\left(L\left(N_{\mathrm{ph}}|0\right)P_{0}+L\left(N_{\mathrm{ph}}|1\right)P_{1}\right)}.$$

This success rate allows estimation of the fraction of the correctly identified cells containing a biomarker if the flow cytometer setup is followed by a cell sorter and when threshold $N_{\mathrm{ph}}^{*}$ for the number of detected photons is set for each trial. On the other hand, $1-S(N_{\mathrm{ph}}^{*})$ gives the fraction of incorrectly identified cells that do not contain a biomarker. Clearly, for large $N_{\mathrm{ph}}^{*}$, the success rate can be nearly perfect, even in a situation with high background noise. The downside of choosing a high $N_{\mathrm{ph}}^{*}$ is the fraction of biomarkers that will not be identified:$$F=\sum_{N_{\mathrm{ph}}=0}^{N_{\mathrm{ph}}^{*}-1}L\left(N_{\mathrm{ph}}|1\right),$$
which, in the case of rare events, may not be achievable (see red curves in Figure 4a–d). Therefore, the probability *F* to miss a biomarker needs to be considered at the same time as the success rate. We present this statistical analysis for experimental scenarios that have different signal and noise intensities. We kept a constant average signal intensity of eight photons per measurement and varied the noise intensity from 2 to 16, Figure 4.

The graphs presented in Figure 4 show how the success rate varies as the probability of encountering a single emitter changes, depending on the threshold for the number of detected photons. Based on the results of our model, we observed that SNR>4 is required to achieve a success rate above 50% for the occurrence rate below 10${}^{-4}$ and F>50%. We can also see that for lower SNR, the success rate is drastically reduced; for an SNR<0.5, it becomes virtually impossible to detect events with an occurrence probability lower than $10^{-1}$.

The success rate could be further improved by collecting and cross-referencing other information. For instance, if the occurrence of a rare single biomarker event is correlated with a different event, such as a scattering burst, or fluorescence activity at a different wavelength, that extra information can be employed. We did not investigate this procedure here, because such cross-referencing is typical for traditional flow cytometry analysis [28].

A similar procedure can be followed to establish the success identification rate and for discerning between zero and N>1 emitters. In some cases, it may be desirable to distinguish between ensembles with a low and a high number of emitters. For instance, the measurement of the over-expression of the HER-2 gene is used to diagnose breast cancer, whereby the case with N=2 HER-2 genes need to be distinguished from N>5 [29,30]. As it can be seen from the Figure 5, discrimination between two and six emitters may be achieved with much lower SNR≳0.5 than single emitter detection for the same signal intensity of $\langle N_{b}\rangle =8$ photons per emitter per trial.

## 5. Conclusions

In this work, we discussed a detection scheme and its calibration for flow cytometry based on quantum measurements. In particular, we discuss how the identification of single-photon emitters can be achieved and verified from first principles of quantum optics, the second-order autocorrelation measurement. We show that using these first principles, the in situ properties of signal and noise can be derived. This method establishes a photonic scale that provides number-of-emitters-resolving capability in flow cytometry and therefore the absolute self-calibration of the flow cytometer. The scheme proposed here is robust to photon loss since the second-order correlation function is loss-insensitive.

We also analyzed the performance of a flow cytometer at its limit of discriminating rarely occurring single biomarkers from random photon bursts due to noise and showed that it can be performed with the high success rate for practically achievable SNR values >4. In addition, by analyzing the problem of discrimination of overexpressed HER-2 gene, we showed that successful cell sorting is possible for much lower SNR>0.5. The capacity to detect a single emitter in flow cytometry and to distinguish between different number of emitters (biomarkers) can be an extremely advantageous tool for the diagnosis of early-stage diseases or the monitoring of a disease’s recurrence. To substantiate the feasibility of our scheme, we conducted fluorescence measurements to estimate an achievable SNR. For this measurement, we used a 400 nm pulsed laser and commercially available biomarkers (Qdot nanocrystals, Life Technologies, Thermo Fisher Scientific, Waltham, MA, USA), at a concentration of approximately one biomarker in the detection volume of a microscope objective with a numerical aperture of 0.9. The manufacturer’s provided concentration was used to determine the appropriate dilution. To measure the noise, we used a blank sample of distilled water. By these means, we obtained an SNR of 2.7, which led us to conclude that the scheme proposed in this work is feasible with present-day technology and instrumentation.

## Figures and Tables

**Figure 1 sensors-22-01136-f001:**
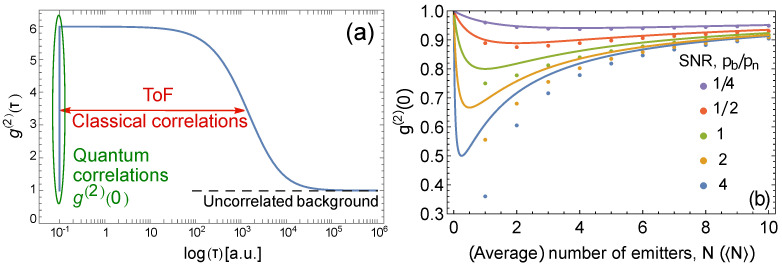
(**a**) Typical dependence of $g^{\left(2\right)}$ on the correlation delay time, τ. The dip at τ=0 is due to non-classical correlations. The shoulder is due to intensity correlations that can be classical. These correlations occur for $0<\tau \ll t_{\mathrm{ToF}}$, where $t_{\mathrm{ToF}}$ is the time of flight. At longer τ, the uncorrelated background emerges. (**b**) $g^{\left(2\right)}\left(0\right)$ vs. (average) number of emitters in the interrogation volume. The different colors correspond to different values of SNR, $p_{b}/p_{n}$. The circles correspond to the static case common to static fluorescense measurements, where $g^{\left(2\right)}\left(0\right)$ is plotted against the exact number of emitters *N*, Equation (Equation 4). The solid lines correspond to the dynamic case common to flow cytometry measurements, where $g^{\left(2\right)}\left(0\right)$ is plotted against the average number of emitters 〈N〉, Equation (Equation 5). The same horizontal axis is used for both *N* and 〈N〉.

**Figure 2 sensors-22-01136-f002:**
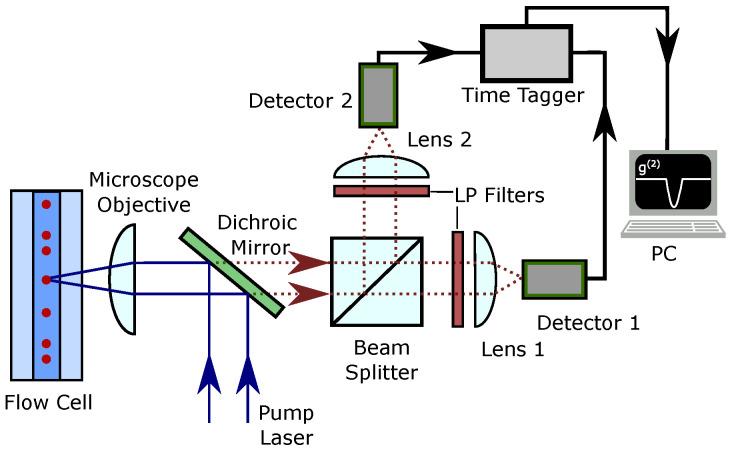
Proposed experimental setup for autocorrelation measurement on the fluorescence signal emitted by the biomarkers in a flow cytometer.

**Figure 3 sensors-22-01136-f003:**
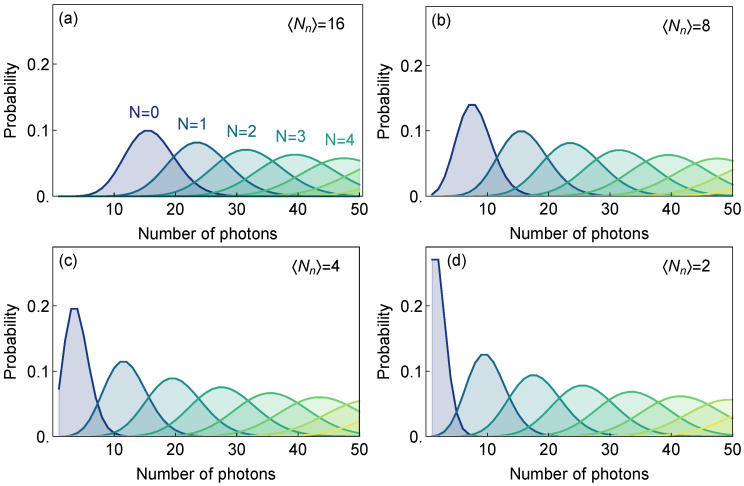
(**a**–**d**) Probabilities $L(N_{\mathrm{ph}}|N)$ to detect a different number of photons in a single trial (consisting of approximately 2400 excitation pulses at 80 MHz repetition rate) of the ensemble with given number of emitters *N* for the average signal intensity $\langle N_{b}\rangle =8$ and set of different noise levels $\langle N_{n}\rangle =\left\{16,8,4,2\right\}$ (SNR = {0.5,1,2,4}).

**Figure 4 sensors-22-01136-f004:**
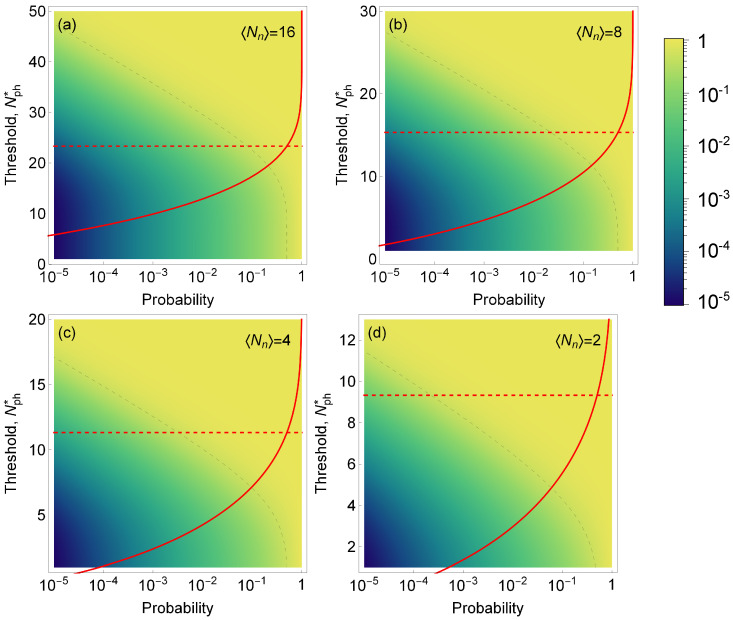
Color map of the success rate *S* (i.e., the fraction of time when the volume identified as containing a single emitter indeed contains a single emitter) as a function of the probability of encountering a single emitter in a sample (horizontal axis) and the threshold for number of photons detected (left vertical axis). Dashed gray line is the isoline for a success rate of 50%. Red line: the probability *F* of erroneously discarding trials containing a biomarker as function of the threshold level $N_{\mathrm{ph}}^{*}$; dashed red line shows $N_{\mathrm{ph}}^{*}$ for which $F(N_{\mathrm{ph}}^{*})=50$%. We assume a fixed signal intensity $\langle N_{b}\rangle =8$ (photons per trial) and noise intensities $\langle N_{n}\rangle =\left\{16,8,4,2\right\}$ in (**a**–**d**), respectively. Note, that even though the density plot and red curves are shown on the same plot, the horizontal axes have different meaning for *F* and *S*.

**Figure 5 sensors-22-01136-f005:**
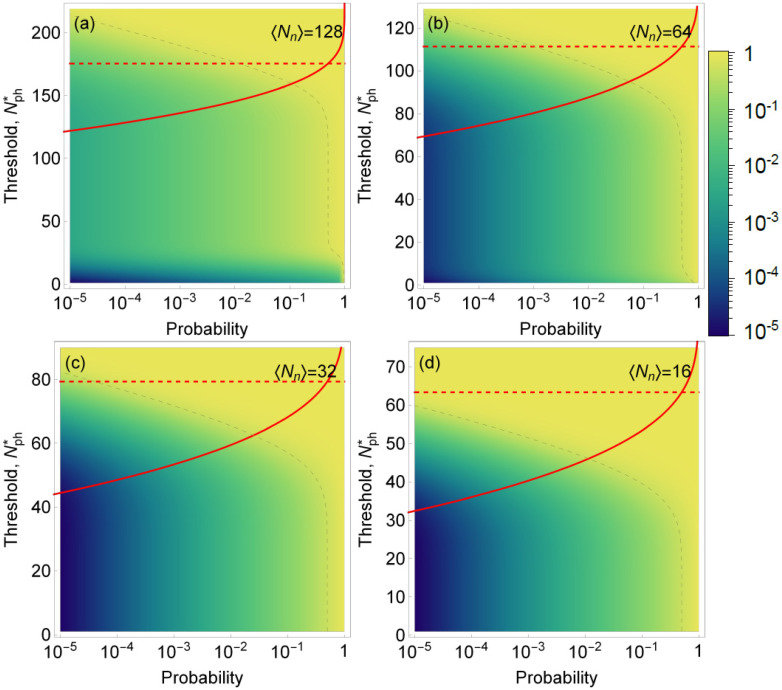
Color map of the success rate *S* (i.e., the fraction of time when the volume identified as that containing six emitters indeed contains six emitters as opposed to just two emitters) as a function of the probability of encountering a volume with six emitters in a sample (horizontal axis) and the threshold for number of photons detected (left vertical axis). Dashed gray line is the isoline for success rate of 50%. Red line: the probability *F* to erroneously discard trials containing six biomarkers as a function of the threshold level $N_{\mathrm{ph}}^{*}$; dashed red line shows $N_{\mathrm{ph}}^{*}$ for which $F(N_{\mathrm{ph}}^{*})=50$%. This example models the identification of overexpressed HER-2 gene with six copies in a cell versus normal two copies (six biomarkers versus two biomarkers). We assumed a fixed signal intensity per biomarker $\langle N_{b}\rangle =8$ (photons per trial) and noise intensities $\langle N_{n}\rangle =\left\{128,64,32,16\right\}$ in (**a**–**d**), respectively. Note that even though the density plot and red curves are shown on the same plot, the horizontal axes have different meanings for *F* and *S*.

## Data Availability

Not applicable.

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
