# Peer review of "In Situ Flow Cytometer Calibration and Single-Molecule Resolution via Quantum Measurement"

_sensors, 2022, doi:10.3390/s22031136_

Round 1

Reviewer 1 Report

Based on first principles of quantum optics, the authors have theoretically investigated how the identification of single-photon emitters can be achieved in flow cytometers. The main idea was to derive a theoretical model for sensing biomarkers at the single-molecule level based on the measurement of the second-order correlation function g2(t) of their emitted light. To this end, the authors have considered realistic experimental scenarios where the biomarker signal would be polluted by noise in a faint coherent light field. Using this assumption, they have obtained an approximate expression for the g2(0) function in terms of the average number of emitters and the probabilities of detecting one photon from a noise source or from a single biomarker. By this expression, in situ properties of signal and noise specific to an experimental setup can be derived from the measurement of g2(0) for a known number of emitters.

Overall, the manuscript seems technically correct and presents a very nice theoretical study on single-photon detection in flow cytometers based on the measurement of intensity-intensity correlation functions. More importantly, the theoretical framework presented by the authors may enable absolute calibration of flow cytometers based on quantum measurements and could be of great interest to biomedical applications such as diagnosis and monitoring of diseases.

In conclusion, since the present manuscript meets the quality standards of Sensors, this reviewer does recommend its acceptance for publication as it is.

Author Response

We appreciate the reviewer's time and insightful comments on our
manuscript.

Reviewer 2 Report

The manuscript by Sabines-Chesterking et al provides a framework to draw a detection baseline for single emitters and enable absolute calibration of a flow cytometer based on quantum measurements. 

The manuscript would be greatly enhanced with some proof of principle experiments rather than being purely theoretical. There will be differences in performance of individual flow cytometers and so it would be highly useful to carry out a multicentre study utilising the same samples in which the concentrations of each marker are known. In addition, this should be backed up with analyte analysis using an alternative and well characterised methodology.

Author Response

"The manuscript would be greatly enhanced with some proof of principle experiments rather than being purely theoretical. There will be differences in performance of individual flow cytometers and so it would be highly useful to carry out a multicentre study utilising the same samples in which the concentrations of each marker are known. In addition, this should be backed up with analyte analysis using an alternative and well characterised methodology."

We point out that this manuscript aims at presenting our theoretical model to motivate a future experiment. We agree with the reviewer's comments that the motivation will be significantly stronger with an experimental feasibility study.  We have now included preliminary measurements of the signal-to-noise ratio of an experimental setup similar to the one shown in figure 2 using commercial quantum dots as a sample. The manuscript is modified by adding the following text: 

"To substantiate the feasibility of our scheme, we have conducted fluorescence measurements to estimate an achievable SNR. For this measurement, we used a 400 nm pulsed laser and commercially available biomarkers (Qdot nanocrystals, Life Technologies, Thermo Fisher Scientific) [31], at a concentration of approximately 1 biomarker in the detection volume of the microscope objective with a numerical aperture of 0.9. The manufacturer's provided concentration was used to determine the appropriate dilution. To measure noise, we used a blank sample of distilled water. By these means, we obtained an SNR of 2.7 which leads us to conclude that the scheme proposed in this work is feasible with present-day technology and instrumentation.  "

We believe that a multicentre study and analyte analysis are beyond the scope of the present manuscript.

Reviewer 3 Report

I quite like the idea of this manuscript, which is to use first principles quantum mechanical methods to develop a better method for fluorescence-based detection from flow cytometry of complex samples. Nonetheless, there are a few relatively minor issues that need to be addressed before I can recommend publication. These include:

  1. In the first sentence of the introduction, the authors define flow cytometry as the “most common technique” used for measuring a range of cell properties. This is likely to be hyperbolic and should probably be rephrased, unless the authors can provide supporting evidence that it is in fact the “most common” technique.
  2. In the introduction, the authors refer to the “quantum-mechanical formalism of the second-order coherence.” This phrase needs to be defined, at least briefly, for readers who may be unfamiliar with the concept being referred to herein.
  3. In the introduction, the sentence that starts “making our scheme suitable for the diagnosis of early-stage diseases” is a sentence fragment and needs to be rephrased.
  4. In the introduction, the authors list possible sources of “optical background noise,” including “Raman scattering, autofluorescence, or scattered pump light.” This is far from a complete listing of possible sources of such noise, and the list should probably be expanded to include a broader range of possibilities.
  5. The text that is written within Figure 1 is extremely difficult to read. The authors should provide a new Figure with more readable text.

More generally, even though this is not an experimental paper, it would benefit from some comparison to experimentally obtained results, particularly those that involve flow cytometry detection using currently existing methods, so that the relevance of this option can be more easily assessed. This kind of comparison would directly indicate the proposed benefits of adopting the reported theoretical approach to analyze such systems.

Author Response

We are grateful for the reviewer's comment's which we have addressed below:

"1. In the first sentence of the introduction, the authors define flow cytometry as the “most common technique” used for measuring a range of cell properties. This is likely to be hyperbolic and should probably be rephrased, unless the authors can provide supporting evidence that it is in fact the “most common” technique."

We have amended the manuscript with the following text:

"Flow cytometry is a commonly used technique to measure a wide range of cell properties in a high throughput manner "

"2. In the introduction, the authors refer to the “quantum-mechanical formalism of the second-order coherence.” This phrase needs to be defined, at least briefly, for readers who may be unfamiliar with the concept being referred to herein."

We have expanded the text to include a brief explanation of what second-order coherence refers to:

"these claims have not been rigorously proven using the quantum-mechanical formalism of the second-order coherence, a method that describes intensity correlations of light sources commonly measured using a Hanbury Brown and Twiss interferometer."

"3. In the introduction, the sentence that starts “making our scheme suitable for the diagnosis of early-stage diseases” is a sentence fragment and needs to be rephrased."

We have now written:

This level of sensitivity makes our scheme suitable for the diagnosis of early-stage diseases or the monitoring of disease recurrence.

"4. In the introduction, the authors list possible sources of “optical background noise,” including “Raman scattering, autofluorescence, or scattered pump light.” This is far from a complete listing of possible sources of such noise, and the list should probably be expanded to include a broader range of possibilities."

We have expanded the list of sources of noise

Thus, it is common to encounter experiments plagued with optical background noise, such as thermal background, Raman scattering, autofluorescence, phosphorescence, scattered and unfiltered pump light, ambient light leakage, among others. In addition, other effects, such as detector dark counts also contribute to background noise."

"5. The text that is written within Figure 1 is extremely difficult to read. The authors should provide a new Figure with more readable text."

We have increased the font of the text in the figure to make it easier to read.

"More generally, even though this is not an experimental paper, it would benefit from some comparison to experimentally obtained results, particularly those that involve flow cytometry detection using currently existing methods, so that the relevance of this option can be more easily assessed. This kind of comparison would directly indicate the proposed benefits of adopting the reported theoretical approach to analyze such systems."

We agree with the fact that the manuscript would benefit from experimentally supported results. However, we consider that a full experimental analysis is beyond the scope of the present work. Nonetheless, we have now included measurements of the signal-to-noise ratio of an experimental setup similar to the one shown in figure 2 using commercial quantum dots as a sample. This supports our claims that the SNR needed four our scheme to work is achievable. We incorporated the following text in the conclusions:

"To substantiate the feasibility of our scheme, we have conducted fluorescence measurements to estimate an achievable SNR. For this measurement, we used a 400 nm pulsed laser and commercially available biomarkers (Qdot nanocrystals, Life Technologies, Thermo Fisher Scientific) [31], at a concentration of approximately 1 biomarker in the detection volume of the microscope objective with a numerical aperture of 0.9. The manufacturer's provided concentration was used to determine the appropriate dilution. To measure noise, we used a blank sample of distilled water. By these means, we obtained an SNR of 2.7 which leads us to conclude that the scheme proposed in this work is feasible with present-day technology and instrumentation. "
